# Enhancing Reproducibility and Replicability in Remote Sensing Deep Learning Research and Practice

Aaron E. Maxwell [1,*], Michelle S. Bester [1] and Christopher A. Ramezan [2]

1    Department of Geology and Geography, West Virginia University, Morgantown, WV 26505, USA
2    John Chambers College of Business and Economics, West Virginia University, Morgantown, WV 26505, USA
*    Correspondence: aaron.maxwell@mail.wvu.edu

**Abstract:** Many issues can reduce the reproducibility and replicability of deep learning (DL) research and application in remote sensing, including the complexity and customizability of architectures, variable model training and assessment processes and practice, inability to fully control random components of the modeling workflow, data leakage, computational demands, and the inherent nature of the process, which is complex, difficult to perform systematically, and challenging to fully document. This communication discusses key issues associated with convolutional neural network (CNN)-based DL in remote sensing for undertaking semantic segmentation, object detection, and instance segmentation tasks and offers suggestions for best practices for enhancing reproducibility and replicability and the subsequent utility of research results, proposed workflows, and generated data. We also highlight lingering issues and challenges facing researchers as they attempt to improve the reproducibility and replicability of their experiments.

**Keywords:** deep learning; replicability; reproducibility; semantic segmentation; object detection





## 1. Introduction

Deep learning (DL) has many applications for analyzing geospatial and remotely sensed data including scene classification, pixel-level classification (i.e., semantic segmentation), object detection, instance segmentation (i.e., detecting and differentiating unique instances of a class), pan sharpening, restoration and denoising, and gap removal [1,2]. More specifically, DL methods based on convolutional neural networks (CNNs) have shown great promise and application in remote sensing and other fields, such as computer vision, autonomous vehicles, and medical imaging, in which data can be represented as multidimensional arrays [1–6]. CNNs can characterize patterns by learning weights associated with kernels or moving windows at multiple scales, allowing for the modeling of spatial context, via two-dimensional convolution. More generally, patterns in other dimensions of interest, such as spectral signatures for hyperspectral data or temporal patterns for a time series can be explored using one-dimensional convolution [1,2,6]. Furthermore, other than optical imagery, CNN-based DL has been applied to other forms of remotely sensed data or digital cartographic representations, such as synthetic aperture radar (SAR) backscatter data (e.g., [7–9]), light detection and ranging (LiDAR) point clouds (e.g., [10–12]), raster-based digital terrain variables (e.g., [13–15]), images collected with ground-based or handheld sensors and cameras (e.g., [16,17]), and historic maps (e.g., [18]). For a review of DL applications in remote sensing, see Yuan et al. [19]. Just as traditional machine learning (ML) methods (e.g., shallow artificial neural networks (ANNs), support vector machines (SVMs), decision trees (DTs), and ensemble DTs) superseded parametric methods (e.g., Gaussian maximum likelihood) [20], DL methods are poised to replace ML methods as the dominant techniques for image classification and feature extraction tasks, especially when spatial, spectral, and/or temporal patterns enhance the separability of classes or the differentiation of features of interest from the background [1,2,6,19].

Traditionally, a dominant area of DL research in the field of remote sensing has been augmenting existing algorithms and/or assessing the effectiveness of algorithm architectures and trained models for making predictions or generalizing to new data and/or geographic extents (e.g., [5,14,18,21–23]), as measured using accuracy assessment metrics and comparison to withheld testing data. However, reproducibility and replicability of research results have been less of a focus. Here, we adopt the definitions of reproducibility and replicability defined by the National Academies of Sciences, Engineering, and Medicine [24]. Reproducibility relates to "computational reproducibility—obtaining consistent results using the same input data, computational methods, and conditions of analysis." In contrast, replicability relates to "whether applying the same methods to the same scientific question produces similar results" [24]. In the field of remote sensing specifically, Small [25] highlighted issues of reproducibility and replicability, alongside applicability and interpretability, as grand challenges facing the discipline as these issues reduce the utility of published studies. More generally, Kedron et al. [26] highlighted issues of reproducibility and replicability in geospatial research, whereas Goodchild et al. [27] explored these issues in the broader context of geographic research [27].

Despite promising results and advancements provided by CNN-based DL in remote sensing research and application, reproducibility and replicability issues arise for several reasons, as described below. This communication does not offer an analysis of reporting standards and practices in published studies. Instead, our goal is to outline and discuss the sources of reduced reproducibility and replicability in CNN-based DL research and workflows, with a focus on semantic segmentation, object detection, and instance segmentation, to foster consideration of these issues by researchers and analyst as they undertake research, data generation, documentation and reporting, and academic publication. Moreover, we offer suggestions towards best practices for enhancing reproducibility and replicability; improving the practical and long-term utility of algorithms, trained models, and datasets; and contributing to the advancement of DL applications in remote sensing. We draw from best practices and standards within the fields of remote sensing and geospatial modeling as well as prior recommendations from other fields, such as medical imaging [28,29], computer vision [30], and software engineering [31]. Lastly, we highlight lingering issues and challenges facing researchers as they attempt to improve reproducibility and replicability of experiments and workflows.

## 2. Issues and Needs

It is important to consider the preprocessing operations and pipelines applied to remotely sensed data prior to providing them as input to the modeling process. This includes processing that is performed by the data originator and also the analysts or researchers undertaking the modeling process. Example preprocessing operations often performed on multispectral imagery include georeferencing, orthorectification, coordinate system transformations, contrast enhancements, conversion to radiance or reflectance, resampling or spatial degrading, pan sharpening, and calculation of spectral and/or spatial enhancements. Other forms of remotely sensed data, such as SAR and LiDAR, often undergo complex and unique preprocessing routines [32–34]. We argue that data preparation tasks routinely implemented in remote sensing are often more complex than those applied to more commonly used data, such as true color images collected with handheld cameras or mobile devices.

Traditional ML algorithms generally have less varied and customizable implementations in comparison to CNN-based DL methods. For example, ensemble DT methods, such as random forests (RF), can generally be described by reporting (1) the computational environment and associated software version information used to deploy the algorithm (e.g., commercial software, Python libraries, R packages, etc.), (2) arguments provided to algorithm parameters or ranges of values provided to tuning processes, (3) the hyperparameter tuning methods deployed, and (4) the selected hyperparameters used in the final model training process [35,36]. In comparison to more deterministic methods, such as

linear regression, even if provided with the same training data and feature space, many ML algorithms will yield different final models due to their stochastic nature [35,36]. However, reproducibility can be enhanced by setting a random seed, which allows the stochastic components of the model to remain consistent (i.e., deterministic). This produces the same results when the training process is re-executed, even on a different computer [35–37].

Fully describing and documenting a DL algorithm is more complex than requirements for traditional ML methods for several reasons, as described in Figure 1. First, the architectures can be intricate with many convolutional blocks and associated components and settings that can be manipulated by the analyst or researcher [1–4]. For example, architectures can be augmented to adjust the number of convolutional blocks, the number of operations in each block and the associated number of kernels learned, stride of convolutional operations, means to reduce the size of the data array between convolutional operations (e.g., max pooling), methods to reduce overfitting (e.g., dropouts [38] and batch normalization [38–40]), and activation functions applied [1–4]. To train the algorithm, the analyst must select an optimization algorithm, a learning rate and other optimization algorithm-specific parameters, and a loss function used to guide the training process (e.g., binary cross-entropy). Callbacks and schedulers can be defined to further customize and control the training process; for example, schedulers can be used to alter the learning rate as the algorithm iteratively learns from the training data [1–4]. Thus, increased model and architectural complexity, along with an increased ability to customize the model, although potentially useful for engineering the algorithm to obtain more accurate and/or generalizable results, can reduce reproducibility and replicability. Essentially, the analyst can define an infinite set of architectural manipulations, hyperparameter settings, and training techniques, and conducting a larger number of experiments to assess the impact of settings and modifications on model performance can be untenable due to computational costs. Such issues result in the DL model development processes being complex, difficult to perform systematically, and challenging to fully document.

As mentioned above, the stochastic nature of traditional ML implementations can be controlled by setting a random seed value, allowing for reproducible results. However, this process is currently more complex for DL algorithms. For example, Alahmari et al. [37] explored the reproducibility of DL results for both a scene labeling and a semantic segmentation problem and compared implementations using TensorFlow 1.14 [41]/Keras 2.3 [42] with implementations using the PyTorch 1.3 [43] DL library. They found that reproducibility could be obtained using PyTorch; however, multiple random seeds had to be set along with fixing the CUDA deterministic flag. The CUDA toolkit and cuDNN library allow for DL to be implemented and accelerated on certain graphics processing units (GPUs) [44]. For the TensorFlow/Keras implementation, reproducibility was obtained for the scene labeling application but not the semantic segmentation problem [37]. In this case, issues of reproducibility stemmed from uncontrolled variability resulting from rounding up floating point numbers and the ordering of calculations in the large number of mathematical operations performed during the gradient updates [28,37]. Thus, there is added complexity and required user care necessary for obtaining reproducible models, and limitations still exist in some implementations.

There are also concerns specific to the training data and the training process. Here, we use imagery as an example data source. To learn spatial context information, the algorithm must be trained using small image chips instead of entire image extents or individual pixels (Figure 1). The user must determine the size of the image chips (e.g., 128-by-128 pixels vs. 256-by-256 pixels), how to break larger image extents or datasets into image chips, and whether to allow overlap between adjacent chips. Once training chips are created, the analyst will need to determine an appropriate batch size (i.e., how many examples or image chips will be provided to the algorithm as a group before updating the weights), over how many iterations over the entire dataset, or epochs, to allow the algorithm to learn from the training data, and if and how randomly shuffling of samples will be applied.

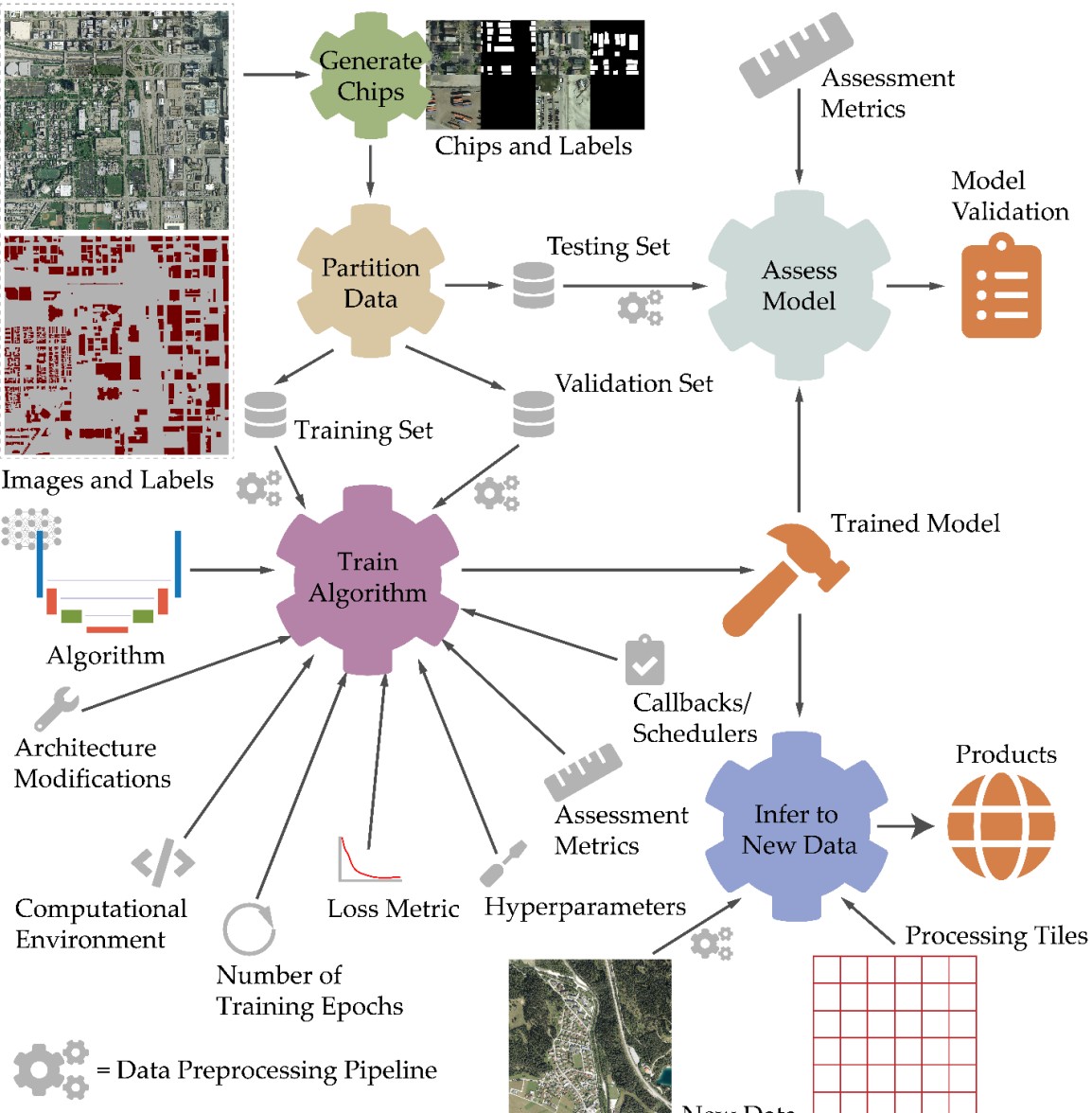

**Figure 1.** Conceptualization of CNN-based deep learning workflow for semantic segmentation of remotely sensed imagery highlighting user-input and customization. Icons from Font Awesome and made available under a CC by 4.0 license.

Moreover, it is generally necessary to determine whether the training process should be stopped early. This is usually accomplished by monitoring the calculated loss for both the training samples and a withheld set of validation samples, which are commonly predicted at the end of each epoch to assess model accuracy and generalization to new data. Monitoring model performance as weights are updated over multiple epochs and determining whether to augment or stop the training process is guided using a user-defined loss metric and, potentially, multiple assessment metrics (e.g., overall accuracy, precision, recall, F1-score, etc.) [19,22,23,45]. Lastly, augmentations of the training data, such as random changes in contrast, sharpening or blurring, rotations, or flips, are often applied to increase the effective number of samples and potentially reduce overfitting to the training data. Such data augmentations require decisions about which augmentations will be used and how often they will be randomly applied [45].

One issue that cannot decrease just the replicability of research findings but also the validity of these findings is data leakage, which occurs when the modeling process inad-

vertently allows the algorithm access to the withheld testing data or information about these data. Since ML and DL methods are known to overfit to the training data, data leakage can cause overly optimistic assessments of model performance [35,36]. Kapoor and Narayanan [46] provide a taxonomy of data leakage causes relating to training and testing models on the same samples, preprocessing using both the testing and training data, performing feature selection using the training and testing data, duplicate samples occurring across data partitions, using inappropriate predictor variables, temporal autocorrelation, nonindependence between training and testing samples, and sampling bias in the testing data [46]. We argue that data leakage can be of particular concern when working with geospatial data due to issues of temporal and/or spatial autocorrelation. For example, Maxwell et al. [23] noted that including overlapping image chips in both the training and testing partitions could lead to overly optimistic assessment. Furthermore, they argue that assessing generalization of models to new data and/or geographic extents requires the generation of testing datasets designed with this purpose in mind [23]. Maggiori et al. [21] offered a good example of a study specifically designed to assess model generalization to new geographic extents.

Another means to potentially improve model performance, especially when the training dataset is not large, and/or to reduce the number of required epochs or time needed to train the model, is to initialize the model using weights learned from another, often very large, dataset, such as the ImageNet [47] or Common Objects in Context (COCO) [48] datasets, a process known as transfer learning [49]. Suppose a model is initialize using such weights, as opposed to a random initialization. In that case, the user can determine which learned parameters or kernel weights will be locked or frozen and which can be updated during the learning process. It is also possible to update weights during only a subset of the training epochs [49].

Similar to traditional ML methods, multiple computational environments can be used to implement DL algorithms and workflows. For example, within the Python [50] data science environment DL can be implemented with Tensorflow [41] or PyTorch [43]. Further, libraries have been developed to simplify the application of DL; for example, Fastai [51], Lightening [52], and Ignite [53] are high-level libraries that simplify the implementation and use of PyTorch. DL has also been integrated into commercial software environments; ArcGIS Pro [54] has implemented several semantic segmentation, object detection, and instance segmentation methods using different backends [54,55]. Given the fast pace of development, libraries can change substantially between builds and releases, resulting in the need to update code developed to support research, workflows, and production pipelines.

At the time of this writing, DL methods are developing quickly with augmentations of existing architectures and methods being proposed in the literature along with new methods that have the potential to drastically impact DL methods and applied research and workflows. Although there were some early developments, the ML algorithms that are most heavily used in remote sensing today (e.g., SVMs, boosted DTs, and RF) developed over several decades, stretching from the early 1980s to mid-2000s [35,36]. In contrast, CNN-based deep learning methods have developed rapidly over the last ten years since the release of AlexNet [56] in 2012. Some of the first techniques developed for semantic segmentation, such as fully convolutional neural networks (FCNs) [57] and UNet [58], date back to only 2014 or 2015. Semantic segmentation methods that have been integrated into the commercial ArcGIS Pro [54,55] software using open-source backends (e.g., Tensorflow and PyTorch) include UNet (2015) [58], PSPNet (2017) [59], and DeepLabv3+ (2018) [60]. Thus, methods developed over the last seven years have already been integrated into commercial geospatial software to support applied mapping tasks. Since traditional ML methods developed at a slower pace, researchers were able to adopt and apply stable, validated, and well-recognized algorithms, such as SVM and RF. Due to the rapid pace of DL advancement, stability in methods to adopt in applied research and mapping tasks has been elusive, further highlighting the need for reproducibility and replicability.

CNN-based methods have generally relied on supervised learning, in which the model must be trained using chips and associated labels [1–4]. However, generating many accurate labels can be expensive, time consuming, or even impossible [61,62]. As a result, semi-supervised learning methods have and are being proposed that allow for the incorporation of unlabeled data into the training process to potentially improve model generalization and/or reduce overfitting to a small training dataset [63–66] Such methods are of specific interest in fields in which unlabeled data can be cheaply and/or easily obtained; for example, in remote sensing, it is generally easier and more efficient to generate many image chips from available date; however, labels are more difficult to generate. Recent advances in the use of transformers in DL have resulted in further progress [67–69]. Thus, the pace of augmentation of existing DL methods and the development of new techniques is unlikely to slow in the near future.

### 3. Recommendations and Best Practices

Prior studies and reviews have noted issues in reporting standards and practices in remote sensing studies. For example, and in the context of accuracy assessment practices, in a review of 100 randomly selected remote sensing DL studies published in 2020, Maxwell et al. [22] documented that only 17% of the papers surveyed included a confusion matrix, reported confusion matrices rarely represented true landscape proportions (i.e., were true population confusion matrices), and that most studies provided inadequate details as to how example data were created and partitioned into separate training, testing and validation datasets. Issues of inadequate reporting of accuracy assessment methods and results have been noted as issues in the remote sensing literature over several decades and prior to the development of DL (e.g., [61,62]). Kapoor and Narayanan [46] documented 329 papers across 17 fields that had methodological flaws that resulted in data leakage and overoptimistic assessment of model performance. Given the complexities and issues described above and highlighted in prior studies, we argue that there is a need to recommend best practices to enhance the long-term viability of data, trained models, and findings and foster reproducibility and replicability in CNN-based DL remote sensing research focused on pixel-level classification, object detection, and instance segmentation. Improvements in transparency and reporting practices is a good first step. The following recommendations are suggested with these goals in mind and for consideration by researchers, authors, reviewers, and journal outlets.

1. Where appropriate and available, data originator, sensor and platform, and product level information should be indicated along with collection date(s) and data identifiers. Pre-processing operations and pipelines should be documented including georeferencing and orthorectification, coordinate transformations, resampling methods, spatial and spectral enhancements, pan sharpening, and contrast enhancements.

2. Whenever possible, code should be made publicly available using an academic or code repository. The code should be well commented and explained, and version numbers of used software, development environments, code libraries, and/or dependencies should be provided. If it is not possible to make code available, then it is of increased importance to clearly document the methods, data, processing, and algorithm architecture in associated research articles and/or other documentation.

3. Specifications for the used hardware should be provided, such as GPU make and model, underlying operating system, and any changes to GPU settings [70,71].

4. Algorithms should be well explained and documented in articles and reports. This should include the base algorithm or architectures used, original citations for the method, and descriptions of any augmentations to the base architecture (e.g., adding batch normalization, using a different activation function, modifying the number of convolutional blocks, or changing the number of learned kernels). Readers should be able to use the described methods to augment the base architecture to reproduce the augmented algorithm used in the study or project.

5. Random seeds can be set to enhance reproducibility. However, needs and methods vary between frameworks, as described by Alahmari et al. [37]. Users should consult documentation to determine how to appropriately set random seeds to obtain deterministic models or reproducible results. However, this may not be possible in all software or framework environments. As an alternative, researchers could run multiple, randomly initialized models, and report the variability in the final model results, which may be especially useful when comparing algorithms, methods, or feature spaces. However, this may not be possible given the computational costs of running multiple model iterations.

6. The entire training process should be clearly described, including input chip size, number of training samples, number of validation samples, batch size, epochs, callbacks and/or schedulers used, optimization algorithm used and associated parameters (e.g., learning rate), and loss metric implemented.

7. Training, validation, and testing data should be provided if possible. The number of available chips and how chips and associated labels were generated from larger image extents should be clearly documented. The methods used to partition the available chips into training, validation, and testing sets should be well explained. The geographic extent of the dataset(s) and source data should be well described and referenced. Any processing applied, such as rescaling pixel values or applying normalization, requires explanation. Other researchers and analyst should be able to reproduce the experimental workflow to obtain the same values and tensors used in the original analysis. If it is not possible to make the training data available, it is still important to clearly document the workflow used since others may need to implement the same methods in order to apply the algorithm to new data.

8. Studies must be carefully designed so as not to introduce data leakage, which can severely impact replicability and even invalidate model assessment results [46]. Researchers should pay special attention to data partitioning methods, issues of temporal and spatial autocorrelation between samples in the data partitions, and not incorporating testing data into preprocessing and feature selection workflows.

9. If transfer learning is used to initialize the model weights, the source of the weights should be explained, such as the image dataset used. Moreover, it is necessary to explain what weights in the model were updated during the training process and which were not. If some weights were only updated during a subset of training epochs, such as those associated with the CNN backbone used to develop feature maps, this should also be explained and documented.

10. All data augmentations or transformations applied to increase the size of the training dataset and potentially reduce overfitting need to be explained including what transformations were used (e.g., blur, sharpen, brighten, contrast shifts, flip, rotate, etc.) and the probabilities of applying them. For increased transparency, augmented data can be written to files and provided with the original chips.

11. Methods used to validate and assess models should be explained such that they can be reproduced. When evaluating models, it is also important that researchers adhere to defendable methods for assessing model performance and the accuracy of products, such as those suggested by Maxwell et al. [22,23].

As discussed above, there are many technical decisions that researchers and analysts must make as they develop research and production pipelines and workflows. Table 1 lists some papers that we have found especially useful in guiding how best to train algorithms, assess results, and combat overfitting. We suggest that these papers be reviewed for specific, technical recommendations. It should be noted that advancements in algorithms and computational methods will likely require augmentation of current best practices and that some researchers disagree as to current best practices and future research needs.

**Table 1.** Example studies that provide specific, technical recommendations relating to training algorithms, assessing models, and combating overfitting.

| Topic | Title | Citation | Year |
|---|---|---|---|
| Selecting a Loss Metric | "Loss odyssey in medical image segmentation" | Ma et al. [72] | 2021 |
| Learning Rate, Schedulers, and Hyperparameters | "Cyclical learning rates for training neural networks" | Smith [73] | 2017 |
| | "A disciplined approach to neural network hyper-parameters: Part 1–learning rate, batch size, momentum, and weight decay" | Smith [74] | 2018 |
| | "Demystifying learning rate policies for high accuracy training of deep neural networks" | Wu et al. [75] | 2019 |
| Data Leakage | "Leakage and the reproducibility crisis in ML-based science" | Kapoor and Narayannan [46] | 2022 |
| Accuracy Assessment Best Practices | "Classification assessment methods" | Tharwat [76] | 2020 |
| | "Accuracy assessment in convolutional neural network-based deep learning remote sensing studies—Part 1: literature review" | Maxwell et al. [22] | 2021 |
| | "Accuracy assessment in convolutional neural network-based deep learning remote sensing studies—Part 2: recommendations and best practices" | Maxwell et al. [23] | 2021 |
| Combating Overfitting | "Batch normalization: Accelerating deep network training by reducing internal covariate shift" | Ioffe and Szegedy [40] | 2015 |
| | "Understanding batch normalization" | Bjorck et al. [39] | 2018 |
| | "A survey on image data augmentation for deep learning" | Shorten and Khoshgoftaar [45] | |
| | "Dropout vs. batch normalization: an empirical study of their impact to deep learning" | Garbin et al. [38] | 2020 |

There are several outstanding issues that reduce the reproducibility and replicability of DL experiments that require further investigation or research. First, there are several practical limitations to undertaking DL research that advances in computer architecture and/or DL methods may aid in minimizing; for example, DL algorithm training processes can be very time-consuming and computationally expensive, limiting our ability to train multiple models to assess variability or to design rigorous, comprehensive experiments that involve many models being trained and/or compared. Such practical limitations can significantly impact experimental designs and the types of questions that can be rigorously investigated. It should be noted that issues arise from the black box nature of DL methods and resulting predictions. Thus, further research on how to visualize and explain these complex models is of great value [77–79].

Musgrave et al. [80] reviewed comparative studies and note methodological flaws, such as failing to attempt to optimize algorithms, which has resulted in unrealistic reported improvements in model performance overtime. This highlights the need to fully document research workflows so that findings can be validated or replicated by others. As noted by Ivie and Thain [81], there is a need to further enhance the reproducibility and replicability of scientific computing in general; however, fundamental, technical, and social barriers still exist and must be overcome. For example, on the subject of geospatial sciences, Tullis and Kar [82] explored the importance of ethically documenting provenance information in studies while also maintaining an individual's location privacy. Thus, there is a need for continued research and tool development to foster reproducible scientific computing, including DL workflows, and for fostering adoption of developed standards and tools.

Hartley and Olsson [71] argued for the need for better methods and means to manage the large volumes of data needed to train DL models; publish or maintain workflow

metadata; and distribute and link algorithm architectures and model weights, data, and associated metadata. Kapoor and Narayanan [46] argued in favor of the use of model info sheets and associated reporting standards for detecting and minimizing data leakage. Sethi and Gil [30] argued for the further development and adoption of semantic workflows in which the research and development processes are well-documented and both workflows and associated datasets are made available and inherently linked. As an added benefit, adoption of semantic workflows may allow for more consistent experimentation and model implementation, easier deployment and/or extension of models, and wider adoption of DL methods by analysts and researchers that lack expertise in the field. Sun et al. [83] developed a cyberinfrastructure, Geoweaver, for managing and documenting complex geospatial artificial intelligence workflows including data preprocessing, algorithm training and testing, and post-processing of results.

Outside of developing tools to allow for easier tracking and documentation of workflows, we argue that a key component of improving reproducibility and replicability is incorporating the use of such tools into standard professional practices, production pipelines, research workflows, training materials, and higher education curricula. Combating the reproducibility and replicability crisis in scientific studies [84] will require fundamental shifts in how we conduct research and communicate findings. When undertaking our own research, we found that we seldom used a single software tool or computational environment. We tended to select our tools based on access, familiarity, and our own professional judgement as to the best tool for a specific task. Thus, we argue that workflow documentation tools must be able to accommodate both commercial and open-source environments and work across multiple software packages, as opposed to being integrated into a single software environment, Lastly, there is a need to shift away from the focus on published articles as the single product of research. Instead, researchers should view a published manuscript as one component of a larger product consisting of data, code, and well-documented workflows.

## 4. Conclusions

Adopting best practices, workflows, and research methods that enhance reproducibility and replicability is of utmost importance for optimizing the value of data, algorithms, trained models, and research. Given the complexity and customizability of CNN-based DL models, involved data preparation and processing pipelines, and the ability to augment training methods and projects that use these methods face many issues that can reduce or prevent reproducibility and replicability. The rapid advancement of DL methods, technologies, associated tools, and code libraries adds additional hurdles. Thus, we argue for the adoption of the above recommendations and the further investigation of the proposed research needs to optimize the benefits and utility of DL research in remote sensing and the longevity and value of research outcomes.

**Author Contributions:** Conceptualization, A.E.M.; formal analysis, A.E.M.; writing—original draft preparation, A.E.M.; writing—review and editing, A.E.M., M.S.B. and C.A.R. All authors have read and agreed to the published version of the manuscript.

**Funding:** Funding was provided by AmericaView, which is supported by the U.S. Geological Survey under Grant/Cooperative Agreement No. G18AP00077. The views and conclusions contained in this document are those of the authors and should not be interpreted as representing the opinions or policies of the U.S. Geological Survey. Mention of trade names or commercial products does not constitute their endorsement by the U.S. Geological Survey. Funding for this research has also been provided by the National Science Foundation (NSF) (Federal Award ID No. 2046059: "CAREER: Mapping Anthropocene Geomorphology with Deep Learning, Big Data Spatial Analytics, and LiDAR"). Any opinions, findings, and conclusions or recommendations expressed in this material are those of the author(s) and do not necessarily reflect the views of the National Science Foundation.

**Data Availability Statement:** No data were generated in this communication.

**Acknowledgments:** We would also like to thank four anonymous reviewers and the academic editor whose comments strengthened the work.

**Conflicts of Interest:** The authors declare no conflict of interest.

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
