# Peer review of "Enhancing Reproducibility and Replicability in Remote Sensing Deep Learning Research and Practice"

_remotesensing, doi:10.3390/rs14225760_

Round 1

Reviewer 1 Report

This communication comprehensively  discusses key issues associated with convolutional-neural network (CNN)-based DL in remote sensing for undertaking semantic segmentation, object detection. The outcome of this study provides the suggestions for best practices and research needs for enhancing replicability and reproducibility and the subsequent utility of research results. Overall, the topic of this research is interesting, and the manuscript was well organised and written. The detailed comments are provided as follows.

1.       The contribution and innovation of the manuscript should be clarified clearly in abstract and introduction.

2.       Broaden and update the literature review on CNN or deep learning for practical applications, including image processing or semantic segmentation. E.g. Vision-based concrete crack detection using a hybrid framework considering noise effect; Crack detection of concrete structures using deep convolutional neural networks optimized by enhanced chicken swarm algorithm.

3.       A table or graph is suggested to be added for categorising remote sensing deep learning studies.

4.       Challenges of deep learning practice should be discussed.

5.       More future research should be included in conclusion part.

Reviewer 2 Report

This article reported the most well-known issues in training deep learning models to deal with certain segmentation and object detection tasks in remote sensing images. The authors reported a set of best practices for scientific method reproducibility which is necessary to compare different algorithm implementations in an open research context.

The recommendations reported in this communication are very general, and the outcomes of the study can be easily found on the net.

To be published in the form of a communication in this journal (https://www.mdpi.com/journal/remotesensing) the authors should deepen their reflection otherwise I propose that the communication be rather published as it is in the form of a blog article (ex. medium.com)

For example, in the communication the point relating to the question of whether reproducibility is more difficult or not in remote sensing than in other sciences and why? could be investigated. Please refer to this blog https://blog.ml.cmu.edu/2020/08/31/5-reproducibility/

Reviewer 3 Report

In the Introduction section, you discuss reproducibility. See this recent work: https://arxiv.org/pdf/2207.07048.pdf. Explain in detail why this is a significant issue

Essentially, the analyst can define an infinite set of architectural manipulations, hyperparameter settings, and training techniques, resulting in the deep learning model development processes being science, art, and tradecraft - This reads like a hostile statement. Can you elaborate on what you mean by the development of DL as art and tradecraft?

The concerns that you raised in Section 2 are genuine, appreciated and well-known. What I do not understand is what new information you are sharing. It needs to be backed with data, how many studies have failed to address this? What are the critical issues in reproducibility, pick some papers and showcase precisely what information it's missing and what can be done.

At this point, the idea of the manuscript is good but lacks novelty. 

Reviewer 4 Report

This is an interesting paper. There are some suggestions for revision.

1. More technical solutions published in 2022 should be discussed.

2. The importance of this paper should be discussed.

3. More numerical analysis of existing issues should be given.

4. More technical details analysis should be given in Section 3.

5. The comparison of existing solutions should be given. 

6. Overall, the discussion of this manuscript is too superficial. More guidance at technical level should be given.

7. The relationship between issues/recommendations and remote sensing is not clear. It seems the recommendations are not specialized for remote sensing issues.

8. The special features of remote sensing should be discussed.

9. The related simulations/experiments should be given to demonstrate existing issues.

10. As discussed in Section 3 recommendations and best practices, more detailed analysis of the mentioned issues/existing solutions should be given.

Round 2

Reviewer 3 Report

Responses are satisfactory and can be accepted.

Reviewer 4 Report

More techincal details and quantitative analysis should be given. 
